# Learning to Compose Degradations: Degradation-Adaptive Codebook for All-in-One Image Restoratio

## Abstract

All-in-one image restoration aims to develop a single model for diverse degradations, a challenge whose success critically hinges on the precise representation of the underlying degradation process. Existing methods simplify this challenge by mapping each degradation to a coarse-grained, monolithic representation—effectively treating them as discrete categories (e.g., "haze," "noise"). This paradigm, even in prompt-learning variants, fundamentally fails to capture the continuous and fine-grained nature of real-world corruptions, such as varying intensities, leading to suboptimal performance. To address this, we argue that degradations are better represented as a composition of a finite set of learnable, elementary degradation primitives. We introduce DACode, a novel framework built upon a global, learnable codebook embodying these primitives. The core of DACode is a two-stage, dual cross-attention mechanism. First, in the Context-Aware Code Adaptation stage, the codebook primitives act as queries to attend to the input image features, generating a contextually-adapted codebook. Subsequently, in the Code-based Feature Modulation stage, the image features query this adapted codebook, aggregating relevant primitive information to perform targeted feature restoration. This dynamic process allows DACode to construct highly specific restorative features for each input. Notably, our analysis reveals that DACode learns to activate distinct code combinations in response to both varying degradation types (e.g., haze vs. rain) and severities (e.g., light vs. heavy haze), providing direct evidence for its fine-grained modeling capability and interpretability. Extensive experiments show that DACode significantly outperforms state-of-the-art methods across all-in-one restoration benchmarks. Code are availale in an anonymous repository https://anonymous.4open.science/r/DAcode-847A/

## 1 Introduction

Image restoration (Banham & Katsaggelos, 1997) is a fundamental low-level vision task that aims to recover high-quality, clean images from observations degraded by a multitude of factors. While deep learning models have yielded impressive results on specific restoration tasks such as denoising (Zhang et al., 2017), deraining (Li et al., 2018b), and dehazing (Wu et al., 2021), these specialized networks embody a "one-model-per-task" paradigm. This approach is not only resource-intensive but also lacks the flexibility to handle the diverse and often unpredictable degradations encountered in real-world applications. Consequently, developing a single, unified model for All-in-One image restoration has become a key and highly-watched research direction.

Significant progress (Chen et al., 2021) has been made in this area (Li et al., 2022; Park et al., 2023; Guo et al., 2024). A common thread in existing methods, however, is the reliance on a **coarse-grained, categorical representation** of degradation. They simplify the complex problem by mapping each degradation type to a monolithic, high-level label (e.g., "haze"), using mechanisms like task-specific prompts (Potlapalli et al., 2023; Kong et al., 2024; Chen et al., 2024) or classifier heads (Conde et al., 2024). This paradigm is fundamentally ill-equipped to capture the fine-grained, continuous spectrum of real-world corruptions. For instance, it struggles to differentiate between a light mist and a heavy fog (intensity differences), a limitation conceptually illustrated in Figure 1(a). Moreover, attempting to address this continuous spectrum by discretizing it—for instance, requiring

a distinct prompt for each specific noise level (*e.g.*, $\sigma = 15, 25, 50$) as seen in methods like PromptIR (Potlapalli et al., 2023)—inevitably leads to a combinatorial explosion. This inherent limitation in modeling rich details severely constrains the performance and generalization of existing models.

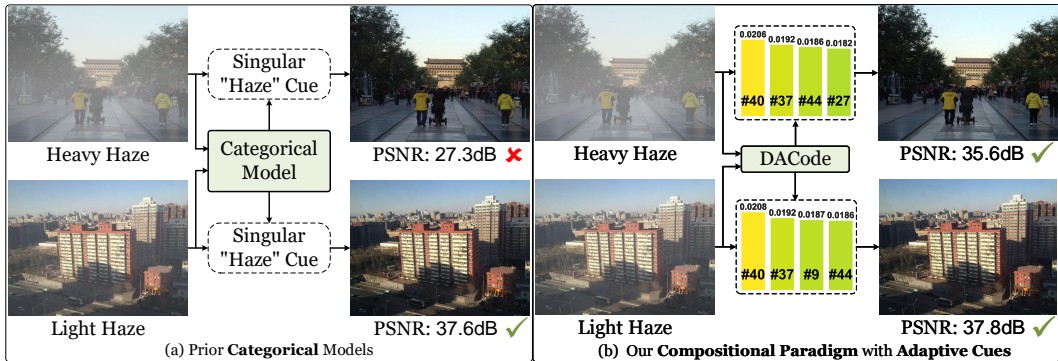

Figure 1: Conceptual illustration of our proposed Compositional Paradigm versus the prior Categorical Paradigm. **(a)** Prior models based on the **Categorical Paradigm**, represented here by the state-of-the-art InstructIR (Conde et al., 2024). This coarse, one-size-fits-all representation leads to suboptimal performance, failing on the challenging case (PSNR: 27.3dB). **(b)** In contrast, our DACode operates under a **Compositional Paradigm**. It intelligently assembles different combinations of fine-grained "degradation primitives" to form bespoke, adaptive cues for each scenario, enabling consistently restoration quality.

To address this fundamental representational bottleneck, we advocate for a paradigm shift towards **fine-grained, compositional modeling**. We posit that any complex degradation can be more effectively represented by adaptively combining a set of learnable, elemental "degradation primitives". Building on this insight, we propose **DACode** Degradation-Adaptive Codebook, a novel framework that materializes this new paradigm, as depicted in Figure 1(b). The core of DACode is a learnable Codebook trained to capture these underlying primitives. Its key innovation lies in a two-stage, dual cross-attention mechanism that forms a bespoke degradation representation for each input. First, the codebook queries the image features to become context-aware. Subsequently, the image features query this adapted codebook to aggregate targeted restorative information. This design empowers our model to transcend discrete categorical constraints and naturally handle continuously varying and mixed degradations.

Our extensive experiments validate the superiority of this approach, with DACode establishing a new state-of-the-art across multiple challenging benchmarks. The main contributions of this work are summarized as follows:

- We identify the limitation of coarse-grained representation in existing methods and propose a new fine-grained, compositional modeling paradigm for all-in-one image restoration.

- We design DACode, a novel framework centered around a degradation-adaptive codebook and a dual cross-attention mechanism, which materializes the proposed paradigm.

- We achieve state-of-the-art performance and provide analyses that validate the model's efficacy and its unique ability for interpretable, fine-grained degradation modeling.

## 2 RELATED WORK

### 2.1 SINGLE-TASK IMAGE RESTORATION

Deep learning-based image restoration has evolved rapidly, beginning with foundational CNN architectures (Dong et al., 2014; Zhang et al., 2017). The performance of these CNN-based models was progressively advanced through the integration of more sophisticated mechanisms, such as channel and spatial attention (Zhang et al., 2018; Zamir et al., 2020), multi-stage refinement schemes (Zamir et al., 2021), and long-range feature interactions (Liu et al., 2018). The advent of

the Vision Transformer (ViT) (Dosovitskiy et al., 2021) marked a significant paradigm shift. Researchers began to tailor Transformer architectures for image restoration to better model long-range dependencies (Yang et al., 2020; Chen et al., 2023b). However, the quadratic computational cost of the self-attention mechanism posed a major bottleneck for high-resolution images. This challenge spurred a wave of innovation focused on efficiency, leading to the development of window-based attention (Wang et al., 2022; Chen et al., 2022), linearized attention mechanisms (Deng et al., 2022), and sparse attention patterns (Chen et al., 2023a). This line of research culminated in powerful and versatile architectures like Restormer (Zamir et al., 2022) and Uformer (Wang et al., 2022), which can achieve state-of-the-art results on various individual restoration tasks. However, despite their architectural unity, these models must be independently trained and stored for each specific degradation. This "one-model-per-task" constraint severely limits their practicality in real-world scenarios that demand handling of multiple, unpredictable degradations, thereby motivating the need for the true All-in-One models discussed in the following section.

## 2.2 All-in-One Image Restoration

Early efforts in all-in-one image restoration, such as IPT (Chen et al., 2021), established the viability of using a single, large Transformer-based backbone for multiple tasks. Following this, a dominant trend has emerged: developing explicit mechanisms to make models degradation-aware. This has been approached through various strategies, including learning degradation representations via contrastive learning (Li et al., 2022), employing explicit classifiers to guide task-specific filters (Park et al., 2023; Hu et al., 2025), and learning distinct task-oriented centers (Zhang et al., 2023). More recently, prompt-based learning has become a popular paradigm. In this approach, a unique prompt is learned for each degradation type to enhance the feature representation of the restoration network, as exemplified by PromptIR (Potlapalli et al., 2023; Kong et al., 2024). This concept has been extended by leveraging textual information for more flexible control (Conde et al., 2024; Luo et al., 2024) or by optimizing the multi-task learning process to resolve conflicts between degradation tasks (Wu et al., 2024). A common denominator across these diverse approaches is their emphasis on distinguishing degradations at a category level. By focusing on high-level categorical distinctions, these methods largely neglect crucial details like varying degradation intensity or texture. Our work, DACode, is fundamentally different. It directly addresses this gap by proposing a compositional approach that models degradations at a much finer granularity, assembling bespoke representations from a learned set of elementary primitives.

## 3 Methodology

Our overall network architecture is a hierarchical U-Net (Ronneberger et al., 2015), as illustrated in Figure 2. The encoder takes a degraded image $I_d \in \mathbb{R}^{H \times W \times 3}$ and passes it through four hierarchical stages. The first three stages each contain a series of Transformer Blocks (TBs) followed by a downsampling layer, which halves the spatial resolution while doubling the feature channel dimension to extract progressively abstract features. The symmetric decoder mirrors this structure with three corresponding upsampling stages, where features from the encoder are re-introduced via skip connections to preserve fine-grained details, ultimately restoring the clean image $I_c \in \mathbb{R}^{H \times W \times 3}$. In line with contemporary all-in-one restorers like AdaIR (Cui et al., 2025) and PromptIR (Potlapalli et al., 2023), the backbone is constructed with Transformer Blocks, whose design is based on the powerful Restormer (Zamir et al., 2022), comprising a Multi-Dconv Head Transposed Attention (MDTA) module and a Gated-Dconv Feed-Forward Network (GDFN), which we refer to as the FFN. The detailed architectures of the MDTA and FFN modules are provided in the supplementary material.

While this powerful backbone excels at modeling spatial context, it lacks a specialized mechanism to adapt to the fine-grained characteristics of diverse degradations, a critical capability for the all-in-one challenge. To address this gap, our key innovation is the strategic insertion of our proposed **Degradation-Adaptive Codebook (DAC) Block**. Following the architectural patterns of recent leading methods (Potlapalli et al., 2023; Cui et al., 2025), we insert a DAC Block within each stage of the decoder. As detailed in Figure 2, each DAC Block is a complete processing unit, comprising our novel **DACode module**—which serves as a degradation-aware attention mechanism—followed

Figure 2: The overall architecture of our proposed network. Our key innovation, the Degradation-Adaptive Codebook (DAC) block, is placed in the decoder stages to guide restoration with fine-grained degradation knowledge.

by the same FFN used in the standard TBs. This design allows our model to inject degradation-specific knowledge at multiple scales during the reconstruction process.

### 3.1 THE DACODE MODULE

The DACode module, detailed in the top panel of Figure 2, is the technical core of our framework and the materialization of our proposed compositional paradigm. It is designed to dynamically generate a bespoke representation for any given degradation through a two-stage, dual cross-attention process: (a) Context-Aware Code Adaptation and (b) Code-based Feature Modulation. Let the input feature map to the module be $X \in \mathbb{R}^{H \times W \times D}$. For processing, we flatten its spatial dimensions to obtain $X_{\text{flat}} \in \mathbb{R}^{N \times D}$, where $N = H \times W$.

At the heart of our framework lies a globally shared and learnable codebook, denoted as $C \in \mathbb{R}^{N_c \times D}$. This codebook consists of $N_c$ code vectors, which we conceptualize as "degradation primitives", each with dimension $D$.

#### 3.1.1 STAGE 1: CONTEXT-AWARE CODE ADAPTATION

For the universal primitives to be effective, they must first be tailored to the context of a specific degraded image. We achieve this by having the primitives *query* the image content via a cross-attention mechanism, where the codebook vectors $C$ serve as queries ($Q$) and the input image features $X_{\text{flat}}$ act as keys ($K$) and values ($V$). This allows the codebook to "read" the image and produce an update vector $\Delta C$:

$$\Delta C = \text{Attention}(\text{LN}(C), X_{\text{flat}}, X_{\text{flat}}),\tag{1}$$

where $\text{LN}(\cdot)$ denotes Layer Normalization. The original codebook is then refined through a residual connection, controlled by a learnable, per-primitive scaling factor $\alpha_c \in \mathbb{R}^{N_c \times 1}$:

$$C' = C + \alpha_c \odot \Delta C.\tag{2}$$

Here, $\odot$ denotes broadcasted element-wise multiplication. This process yields a contextually-adapted codebook $C' \in \mathbb{R}^{N_c \times D}$ that is now conditioned on the specific content and degradation style of the input.

#### 3.1.2 STAGE 2: CODE-BASED FEATURE MODULATION

With the context-aware codebook $C'$ obtained, the second stage performs feature modulation by reversing the roles in the attention mechanism. This time, the image features $X_{\text{flat}}$ serve as queries ($Q$) to the adapted codebook $C'$, which acts as both keys ($K$) and values ($V$). This allows each image feature location to "look up" and aggregate the most relevant restorative information from the entire set of adapted primitives. The modulated feature $X_{\text{mod}}$ is computed as:

$$X_{\text{mod}} = \text{Attention}(\text{LN}(X_{\text{flat}}), \text{LN}(C'), \text{LN}(C')).\tag{3}$$

Intuitively, this step materializes the compositional principle: the restoration of each pixel is guided by a bespoke representation, synthesized on-the-fly by combining the most relevant degradation

primitives from the adapted codebook. The final output $X_{mod}$ is then passed to the subsequent FFN within the DAC Block.

## 4 EXPERIMENT

To rigorously evaluate our proposed DACode framework, we conduct a series of comprehensive experiments on the challenging task of all-in-one image restoration. In this section, we first detail our experimental setup, including the datasets and implementation specifics. Subsequently, we present the main quantitative and qualitative results, where we compare DACode against a range of state-of-the-art methods on both 3-task and 5-task benchmarks. Across all tasks, we quantify restoration quality using two standard metrics: the Peak Signal-to-Noise Ratio (PSNR) and the Structural Similarity Index (SSIM). For both metrics, higher scores signify superior restoration performance.

Table 1: Quantitative comparison (PSNR / SSIM) for all-in-one restoration on three tasks. The best results are in **bold**, and the second-best are underlined.

| Method | Dehazing | Deraining | Denoising on BSD68 | | | Average | Params |
|---|---|---|---|---|---|---|---|
| | SOTS | Rain100L | $\sigma = 15$ | $\sigma = 25$ | $\sigma = 50$ | | |
| AirNet (Li et al., 2022) | 27.94 / 0.962 | 34.90 / 0.967 | 33.92 / 0.933 | 31.26 / 0.888 | 28.00 / 0.797 | 31.20 / 0.910 | 9M |
| PromptIR (Potlapalli et al., 2023) | 30.58 / 0.974 | 36.37 / 0.972 | 33.98 / 0.933 | 31.31 / 0.888 | 28.06 / 0.799 | 32.06 / 0.913 | 36M |
| Art-PromptIR (Wu et al., 2024) | 30.83 / 0.979 | 37.94 / 0.982 | 34.06 / 0.934 | 31.42 / 0.891 | 28.14 / 0.801 | 32.49 / 0.917 | 33M |
| InstructIR (Conde et al., 2024) | 30.22 / 0.959 | 37.98 / 0.978 | 34.15 / 0.933 | 31.52 / 0.890 | 28.30 / 0.804 | 32.43 / 0.913 | 16M |
| PromptIR-TUR (Wu et al., 2025) | 31.17 / 0.978 | 38.57 / 0.984 | 34.06 / 0.932 | 31.40 / 0.887 | 28.13 / 0.797 | 32.67 / 0.916 | 33M |
| AdaIR (Cui et al., 2025) | 31.06 / 0.980 | 38.64 / 0.983 | 34.12 / 0.935 | 31.46 / 0.892 | 28.19 / 0.802 | 32.69 / 0.918 | 29M |
| VLU-Net (Zeng et al., 2025) | 30.71 / 0.980 | 38.93 / 0.984 | 34.13 / 0.935 | 31.48 / 0.892 | 28.23 / 0.804 | 32.70 / 0.919 | 35M |
| MoCE-IR (Zamfir et al., 2025) | 31.34 / 0.979 | 38.57 / 0.984 | 34.11 / 0.932 | 31.45 / 0.888 | 28.18 / 0.800 | 32.73 / 0.917 | 25M |
| **Ours (DACode)** | **31.50 / 0.982** | **39.10 / 0.985** | **34.24 / 0.937** | **31.60 / 0.895** | **28.34 / 0.809** | **32.96 / 0.922** | 29M |

Table 2: Quantitative comparison (PSNR / SSIM) for all-in-one restoration on five tasks. Best results are in **bold**, second-best are underlined. Note that for denoising, we report results for $\sigma = 25$ following standard practice in this setting.

| Method | Dehazing | Deraining | Denoising | Deblurring | Low-Light | Average | Params |
|---|---|---|---|---|---|---|---|
| | SOTS | Rain100L | $BSD68_{\sigma=25}$ | GoPro | LOL | | |
| AirNet (Li et al., 2022) | 21.04 / 0.884 | 32.98 / 0.951 | 30.91 / 0.882 | 24.35 / 0.781 | 18.18 / 0.735 | 25.49 / 0.847 | 9M |
| PromptIR (Potlapalli et al., 2023) | 25.20 / 0.931 | 35.94 / 0.964 | 31.17 / 0.882 | 27.32 / 0.842 | 20.94 / 0.799 | 28.11 / 0.883 | 33M |
| Gridformer (Wang et al., 2024) | 26.79 / 0.951 | 36.61 / 0.971 | 31.45 / 0.885 | 29.22 / 0.831 | 22.59 / 0.831 | 29.33 / 0.904 | 34M |
| InstructIR (Conde et al., 2024) | 27.10 / 0.956 | 36.84 / 0.973 | 31.40 / 0.873 | 29.40 / 0.886 | 23.00 / 0.836 | 29.55 / 0.908 | 17M |
| Transweather-TUR (Wu et al., 2025) | 29.68 / 0.966 | 33.09 / 0.952 | 30.40 / 0.869 | 26.63 / 0.815 | 23.02 / 0.838 | 28.56 / 0.888 | 38M |
| AdaIR (Cui et al., 2025) | 30.53 / 0.978 | 38.02 / 0.981 | 31.35 / 0.889 | 28.12 / 0.858 | 23.00 / 0.845 | 30.20 / 0.910 | 29M |
| VLU-Net (Zeng et al., 2025) | 30.84 / 0.980 | 38.54 / 0.982 | 31.43 / 0.891 | 27.46 / 0.840 | 22.29 / 0.833 | 30.11 / 0.905 | 35M |
| MoCE-IR (Zamfir et al., 2025) | 30.48 / 0.974 | 38.04 / 0.982 | 31.34 / 0.887 | **30.05 / 0.899** | 23.00 / 0.852 | 30.58 / 0.919 | 25M |
| **Ours (DACode)** | **31.13 / 0.981** | **39.27 / 0.986** | **31.54 / 0.894** | 29.46 / 0.886 | **23.21 / 0.860** | **30.92 / 0.921** | 29M |

### 4.1 EXPERIMENTAL SETTINGS

### 4.2 IMPLEMENTATION DETAILS

We propose two model variants built upon a 4-level encoder and 3-level decoder U-Net architecture. Our standard model, **DACode**, has a base channel dimension of 48 and uses [4, 6, 6, 8, 6, 6, 12] Transformer blocks across its seven stages. The lightweight **DACode-s** variant reduces the channel dimension to 32 with a shallower block configuration of [4, 6, 6, 8, 2, 4, 8]. The number of primitives in the DACode module is set to $N_c = 64$ for both variants. Models are trained a total of **150 epochs** on two NVIDIA L40 GPUs using the AdamW optimizer (Loshchilov & Hutter, 2017) to minimize the L1 loss. We use a batch size of 12. The learning rate is initialized to $2 \times 10^{-4}$ and decayed to zero via a cosine annealing schedule. Input patches of size $128 \times 128$ are randomly cropped from training images.

**Datasets and Benchmarks.** To comprehensively evaluate DACode, our experiments are structured across three distinct benchmarks that progressively increase in complexity. First, we use a standard **3-task setting** to assess performance on common, isolated degradations. This includes: **(i) Deraining** on the Rain100L dataset Wenhan Yang & Yan (2017); **(ii) Dehazing** on the SOTS indoor

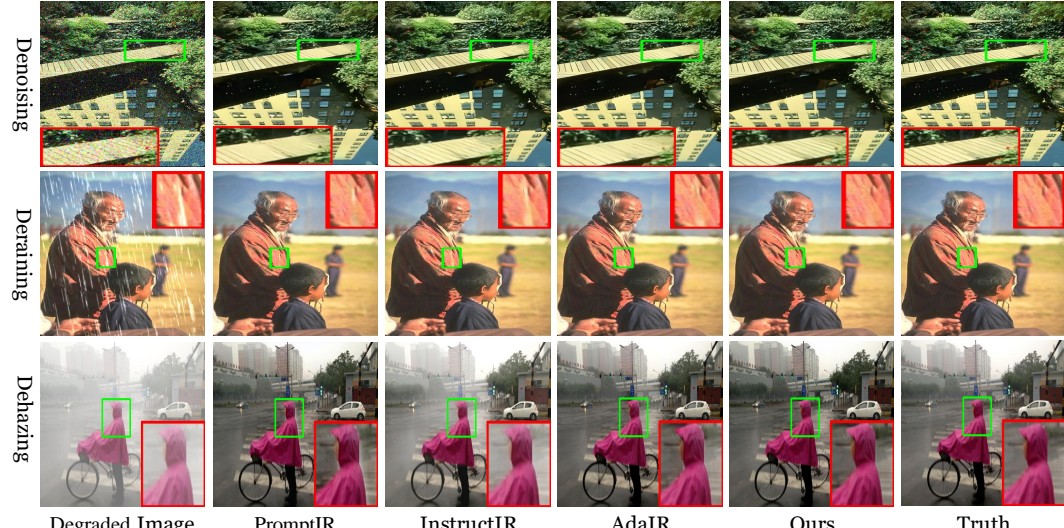

Figure 3: Qualitative comparison for all-in-one restoration on three tasks.

dataset Li et al. (2018a); and **(iii) Denoising**, for which we train on a composite of BSD400 Arbelaez et al. (2010) and WED Ma et al. (2016) with additive white Gaussian noise ($\sigma \in \{15, 25, 50\}$) and test on the BSD68 benchmark Martin et al. (2001). Second, we expand to a more extensive **5-task setting** to demonstrate the model's versatility. This setup supplements the above tasks with **(iv) Deblurring** on the GoPro dataset Nah et al. (2017) and **(v) Low-Light Enhancement** on the LOL-v1 dataset Wei et al. (2018). Finally, to directly validate our model's core capability in handling complex scenarios, we conduct experiments on the **CDD11** dataset Guo et al. (2024) with **Composited Degradations** . This benchmark is specifically designed to test performance on mixed corruptions, containing 11 degradation types that include both single modalities (e.g., rain, snow, haze, low-light) and their combinations (e.g., haze-rain, low-light-snow).

### 4.3 QUANTITATIVE AND QUALITATIVE COMPARISONS

To comprehensively evaluate our framework, we first benchmark DACode against state-of-the-art (SOTA) methods on two standard all-in-one settings: a foundational 3-task benchmark and a more extensive 5-task benchmark. The quantitative results are presented in Table 1 and Table 2. Our DACode framework demonstrates clear superiority, achieving the best overall performance in both settings. On the demanding 5-task benchmark, for instance, DACode surpasses the powerful MoCE-IR (Zamfir et al., 2025) baseline by a notable **0.34 dB** in average PSNR. These quantitative improvements are visually substantiated by our qualitative results in Figure 3. The comparisons reveal our model's enhanced ability to restore fine-grained details and vibrant colors, such as the intricate skin texture in the deraining example and the faithful color rendition of the cyclist's raincoat in the dehazing case.

Beyond the standard benchmarks of isolated degradations, we further probe our model's capabilities in more realistic and complex scenarios using the **CDD11** benchmark for composite degradations. For a fair comparison against the predominantly lightweight methods evaluated on this dataset, we utilize our smaller variant, **DACode-S**. As shown in Table 3, our approach demonstrates overwhelming superiority. Notably, DACode-S achieves an average PSNR of 29.81 dB, surpassing the next-best method, Moce-IR-S, by a substantial margin of **0.76 dB**, despite having a comparable model size. This commanding performance is consistent across all 11 degradation types, spanning single, double, and even the most challenging triple-composite corruptions. This result provides strong, direct evidence for the effectiveness of our fine-grained, compositional paradigm in disentangling and restoring complex, real-world image degradations.

Collectively, these strong quantitative and qualitative results across all benchmarks validate the effectiveness and robustness of our proposed framework.

Table 3: Quantitative comparison on the CDD11 dataset. Best results are in **bold**, second-best are underlined. **(Top)** PSNR results. **(Bottom)** SSIM results.

| Method | Haze(H) | Low(L) | Rain(R) | Snow(S) | H+R | H+S | L+H | L+R | L+S | L+H+R | L+H+S | Average | Params. |
|---|---|---|---|---|---|---|---|---|---|---|---|---|---|
| AirNet (Li et al., 2022) | 24.21 | 24.83 | 26.55 | 26.79 | 22.21 | 23.29 | 23.23 | 22.82 | 23.29 | 21.80 | 22.24 | 23.75 | 8.9M |
| PromptIR (Potlapalli et al., 2023) | 26.10 | 26.32 | 31.56 | 31.53 | 24.54 | 23.70 | 24.49 | 25.05 | 24.51 | 24.49 | 23.33 | 25.97 | 38.5M |
| OneRestore (Guo et al., 2024) | 32.52 | 26.48 | 33.40 | 34.31 | 29.99 | 30.21 | 25.79 | 25.58 | 25.19 | 24.78 | 24.90 | 28.47 | 6.0M |
| Moce-IR-S (Zamfir et al., 2025) | 32.66 | 27.26 | 34.31 | 35.91 | 29.93 | 30.19 | 26.24 | 26.25 | 26.04 | 25.41 | 25.39 | 29.05 | 11.0M |
| **DACode-S (Ours)** | **34.30** | **27.32** | **35.22** | **36.83** | **31.24** | **31.35** | **26.59** | **26.64** | **26.58** | **25.97** | **25.97** | **29.81** | 12.5M |

| Method | Haze(H) | Low(L) | Rain(R) | Snow(S) | H+R | H+S | L+H | L+R | L+S | L+H+R | L+H+S | Average | Params. |
|---|---|---|---|---|---|---|---|---|---|---|---|---|---|
| AirNet Li et al. (2022) | 0.951 | 0.778 | 0.891 | 0.919 | 0.868 | 0.901 | 0.779 | 0.710 | 0.723 | 0.708 | 0.725 | 0.814 | 8.9M |
| PromptIR Potlapalli et al. (2023) | 0.969 | 0.805 | 0.946 | 0.960 | 0.924 | 0.925 | 0.789 | 0.771 | 0.761 | 0.789 | 0.747 | 0.853 | 38.5M |
| OneRestore Guo et al. (2024) | 0.990 | 0.826 | 0.964 | 0.973 | 0.957 | 0.964 | 0.822 | 0.799 | 0.789 | 0.788 | 0.791 | 0.878 | 6.0M |
| Moce-IR-S Zamfir et al. (2025) | 0.990 | 0.824 | 0.970 | 0.980 | 0.964 | 0.970 | 0.817 | 0.800 | 0.793 | 0.789 | 0.790 | 0.881 | 11.0M |
| **DACode-S (Ours)** | **0.991** | **0.834** | **0.974** | **0.981** | **0.968** | **0.972** | **0.831** | **0.814** | **0.808** | **0.806** | **0.805** | **0.890** | 12.5M |

Table 4: Ablation study on the number of code primitives ($N_c$). The case $N_c = 0$ represents our backbone without the DACode module. Best results are in **bold**.

| $N_c$ | 0 | 32 | 48 | **64 (Ours)** | 80 | 96 | 128 |
|---|---|---|---|---|---|---|---|
| PSNR | 31.98 | 32.51 | 32.72 | **32.96** | 32.74 | 32.93 | **32.96** |
| SSIM | 0.909 | 0.914 | 0.918 | **0.922** | 0.918 | 0.920 | 0.921 |

Table 5: Ablation on the Context-Aware Code Adaptation stage. Best results are in **bold**.

| Method | PSNR | SSIM |
|---|---|---|
| DACode w/o Adaptation | 32.71 | 0.914 |
| **Full DACode (Ours)** | **32.96** | **0.922** |

## 5 ABLATION STUDIES

In this section, we conduct a series of targeted ablation studies to rigorously validate our proposed DACode framework. Our investigation is structured to first analyze the impact of the codebook size ($N_c$), which demonstrates the overall efficacy of our module and determines its optimal configuration. We then specifically investigate the criticality of the first stage in our dual-attention mechanism: the Context-Aware Code Adaptation. Finally, we provide in-depth qualitative visualizations that offer direct evidence of our model's fine-grained and adaptive modeling capabilities. Unless otherwise specified, all ablation experiments are conducted on the **3-task** setting.

### 5.1 ANALYSIS ON THE NUMBER OF CODE PRIMITIVES

The number of learnable primitives, $N_c$, in our codebook is a critical hyperparameter that directly influences the model's representational capacity and parameter overhead. To quantify the overall effectiveness of our DACode module and find an optimal configuration, we evaluate the model's performance while varying $N_c$ in the set $\{0, 32, 48, 64, 80, 96, 128\}$. The $N_c = 0$ setting serves as a crucial baseline, as it effectively removes the DACode module.

The quantitative results, summarized in Table 4, lead to two primary conclusions. First, the results reveal the substantial impact of our DACode module. Removing it entirely ($N_c = 0$) causes a drastic performance drop of 1.83 dB in average PSNR (from 32.96 dB to 31.13 dB). This clearly demonstrates that our proposed module is essential for high-quality restoration. Second, we observe a consistent trend of performance improvement as $N_c$ increases from 32 to 64, with both PSNR and SSIM metrics reaching their peak at $N_c = 64$. Interestingly, further increasing the codebook size beyond 64 leads to performance saturation. While the PSNR at $N_c = 128$ matches our peak result, it offers no additional benefits, incurs a higher parameter cost, and results in a slightly lower SSIM score. This analysis indicates that $N_c = 64$ provides the optimal trade-off between representational capacity and model efficiency. Consequently, we adopt $N_c = 64$ as the default configuration for all our experiments.

### 5.2 EFFICACY OF CONTEXT-AWARE CODE ADAPTATION

We hypothesize that a static codebook of universal primitives is suboptimal for the diverse nature of image degradations. To this end, our DACode module incorporates a crucial first stage: **Context-Aware Code Adaptation**, designed to dynamically refine these primitives based on the input image's context. To validate the importance of this stage, we design an ablation variant named

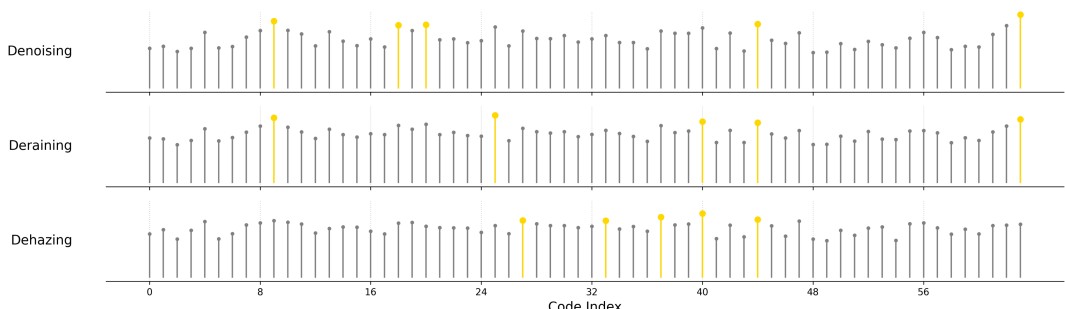

(a) Activation Fingerprints of the codebook across the three restoration tasks.

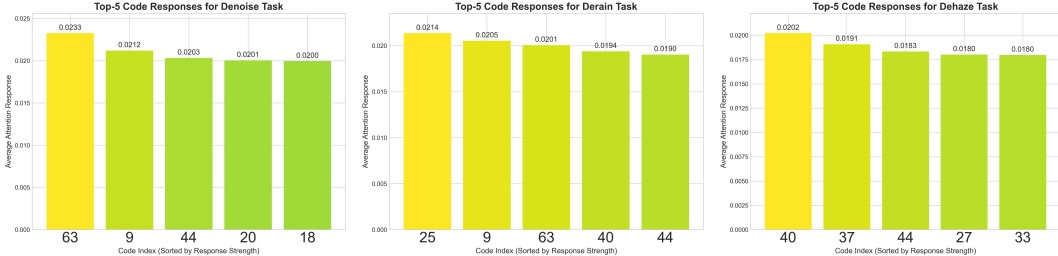

(b) Top-5 most activated primitives for each task, sorted by response strength.

Figure 4: Functional specialization of primitives across different tasks. The visualizations in (a) the Activation Fingerprints and (b) the Top-5 primitives reveal a clear spectrum of specialization. We can identify three primary roles: (1) **Task-Specific Primitives**, which are dominantly activated for a single task (e.g., primitives **#40** and **#37** for Dehazing). (2) **Property-Specific Primitives**, which are co-activated for tasks with shared underlying properties, such as the additive, high-frequency nature of Denoising and Deraining (e.g., primitives **#9** and **#63**). (3) **Universal Primitives**, such as **#44**, which are consistently activated across all three tasks, likely modeling fundamental restoration properties. This clear division of labor validates the richness and interpretability of our compositional approach.

"**DACode w/o Adaptation**", where this mechanism is disabled. Specifically, we bypass the code update step in Eq. 2, forcing the model to use the same static, global codebook for all inputs.

The results, presented in Table 5, confirm our hypothesis. Disabling the adaptation mechanism leads to a substantial performance drop of **1.79 dB** in average PSNR and **0.013** in average SSIM. This significant gap demonstrates that while a global codebook can learn generic primitives, the ability to dynamically specialize them for each unique degradation instance is critical for achieving high-fidelity restoration. Without this context-aware adaptation, the model is constrained to a less effective, one-size-fits-all approach.

### 5.3 QUALITATIVE ANALYSIS: VERIFYING ADAPTIVE MODELING

Beyond quantitative metrics, it is crucial to qualitatively verify that our DACode module operates according to our core motivation. To this end, we visualize the codebook's activation patterns in response to different degradation tasks on their respective standard benchmarks, as presented in Figure 4 and Figure 5. The complete response figures for different tasks are available in the supplementary material.

**Functional Specialization of Primitives.** Figure 4 illustrates how the primitives in our codebook have specialized for different roles when tested on distinct tasks. We observe a clear spectrum of specialization: First, some primitives are highly task-specific. For instance, primitives like **#40 and #37** are dominantly activated only for Dehazing, demonstrating that the model learns representations dedicated to a single degradation family. Second, other primitives specialize in shared properties across tasks. A notable overlap exists between Denoising and Deraining, which share



(a) Denoising: $\sigma = 15$ vs. $\sigma = 50$     (b) Deraining: Light vs. Heavy     (c) Dehazing: Light vs. Heavy

Figure 5: **Severity-aware activation of the DACode module.** The model assembles different combinations of Top-5 primitives in response to varying degradation intensity. This is demonstrated across three tasks: **(a) Denoising**, where the dominant primitive shifts from #41 ($\sigma = 15$) to #32 ($\sigma = 50$); **(b) Deraining**, with a shift from #39 (light) to #61 (heavy); and **(c) Dehazing**, shifting from #42 (light) to #53 (heavy). This behavior validates our model's ability to perform fine-grained modeling by composing different "recipes" of primitives to match the input's specific characteristics. Zoom in for best view.

key primitives such as **#9 and #63**. We attribute this to the shared nature of their data synthesis as **additive degradations** that introduce high-frequency artifacts, suggesting these primitives function as general-purpose high-frequency artifact removers. Finally, we find evidence of universal primitives. **Code #44**, for example, appears as a Top-5 activated primitive across all three tasks. We hypothesize that such universal codes learn to model fundamental, task-agnostic properties essential for general image reconstruction, such as restoring fine textures. This clear division of labor—from universal, to property-specific, to task-specific functions—powerfully demonstrates the richness of the representations learned by our compositional paradigm.

**Severity-Aware Activation.** Furthermore, our model demonstrates a remarkable capability for severity-awareness. As illustrated in Figure 5, the model astutely adapts its response even within the same task. It composes different "recipes" of primitives, with visibly different activation strengths and participating codes, to precisely match the intensity of each input image. This, combined with the hierarchical activation, validates that DACode learns a rich, internal language of degradation rather than relying on monolithic labels.

## 6   CONCLUSION

In this paper, we identified a fundamental limitation in existing all-in-one image restoration methods: their reliance on coarse-grained, categorical representations of degradation. To address this, we proposed a paradigm shift towards fine-grained, compositional modeling. Our framework, DACode, materializes this new paradigm through a novel degradation-adaptive codebook and a dual cross-attention mechanism. The superiority of our compositional approach was validated through extensive experiments, where DACode established a new state-of-the-art on multiple challenging benchmarks. More than just performance, our in-depth analyses provided direct evidence for our core hypothesis. We revealed that the learned codebook exhibits a remarkable degree of functional specialization, with primitives ranging from universal to task-specific. This confirms that our model does not merely memorize tasks, but learns a rich, internal language of degradation, assembling different "recipes" of primitives to precisely match the type and severity of each unique corruption.

## 7   THE USE OF LARGE LANGUAGE MODELS (LLMS)

During the preparation of this paper, we employed a Large Language Model (LLM) to assist with improving the language and readability of the text. The primary use of the LLM was for proofreading, including correcting grammatical errors and refining sentence structure to enhance clarity. We confirm that the LLM was not used for research ideation, developing the methodology, conducting experiments, analyzing results, or drawing conclusions. All intellectual contributions and scientific claims are solely those of the authors.

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

# A APPENDIX

## A.1 INTRODUCTION

This supplementary document provides additional details, experiments, and visualizations to complement our main paper. The contents are organized as follows:

- **Section A.2:** A detailed architectural breakdown of the standard Transformer Block used in our backbone, including the specific structures of the Multi-Dconv Head Transposed Attention (MDTA) and the Feed-Forward Network (FFN).

- **Section A.3:** Complete, unabridged visualizations of the codebook activation responses (i.e., the full 64-primitive bar charts), providing a comprehensive view of the results summarized in the main paper.

- **Section A.4:** Additional qualitative results, providing more visual comparisons of our method against state-of-the-art approaches on various restoration tasks.

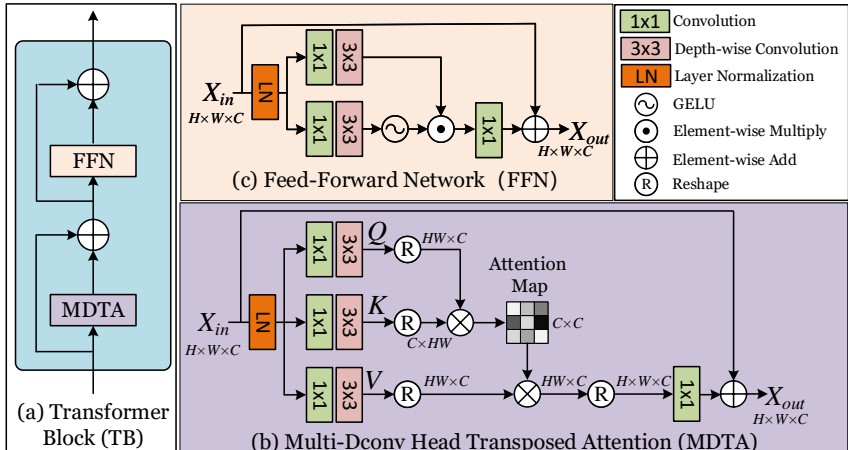

Figure 6: **(a)** Detailed architecture of the core components within a standard Transformer Block (TB). **(b)** The Multi-Dconv Head Transposed Attention (MDTA) module. Unlike standard self-attention which computes spatial relationships, MDTA computes attention across feature channels, making it more efficient for high-resolution images. **(c)** The Gated-Dconv Feed-Forward Network (GDFN), which we refer to as the FFN in our main paper. It employs a gating mechanism to control feature flow and enhance representational power.

## A.2 DETAILED NETWORK ARCHITECTURES

In the main paper, we mentioned that the Transformer Blocks (TBs) in our backbone are based on the Restormer Zamir et al. (2022) design. Each TB is composed of two core components: a Multi-Dconv Head Transposed Attention (MDTA) module for feature aggregation, and a Gated-Dconv Feed-Forward Network (GDFN) for feature transformation, which we refer to as the FFN. Here, we provide a detailed architectural breakdown of these two modules, as illustrated in Figure 6.

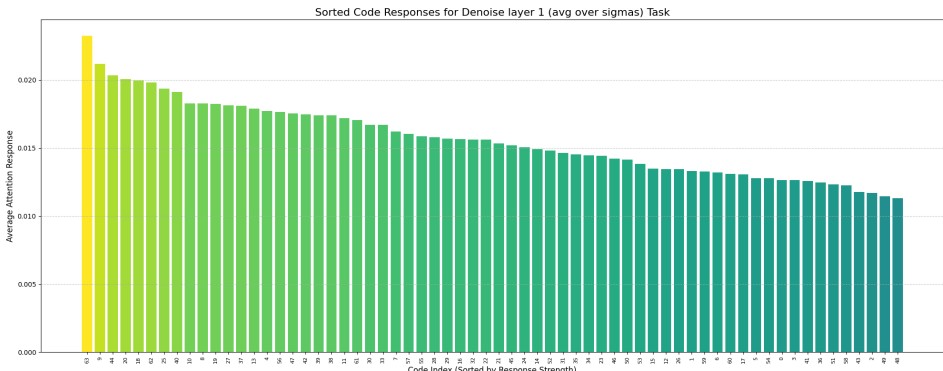

Figure 7: The complete distribution of the 64 code primitives' responses for the **Denoising** task, averaged over all noise levels ($\sigma \in \{15, 25, 50\}$) on the BSD68 dataset.

**Multi-Dconv Head Transposed Attention (MDTA)** The MDTA module, shown in Figure 6(b), is designed to efficiently model long-range dependencies while maintaining a low computational cost. Given an input feature map $X_{in} \in \mathbb{R}^{H \times W \times C}$, it first passes through a Layer Normalization (LN). Then, to generate the query ($Q$), key ($K$), and value ($V$) projections, the normalized features are processed by three parallel branches. Each branch consists of a $1 \times 1$ convolution to adjust the channel dimension, followed by a $3 \times 3$ depth-wise convolution to aggregate local spatial context.

The key innovation of MDTA is its use of transposed attention. Instead of computing attention across the spatial dimension ($N \times N$, where $N = H \times W$), it computes attention across the channel dimension. To achieve this, the key projection $K$ is transposed and multiplied with the query projection $Q$

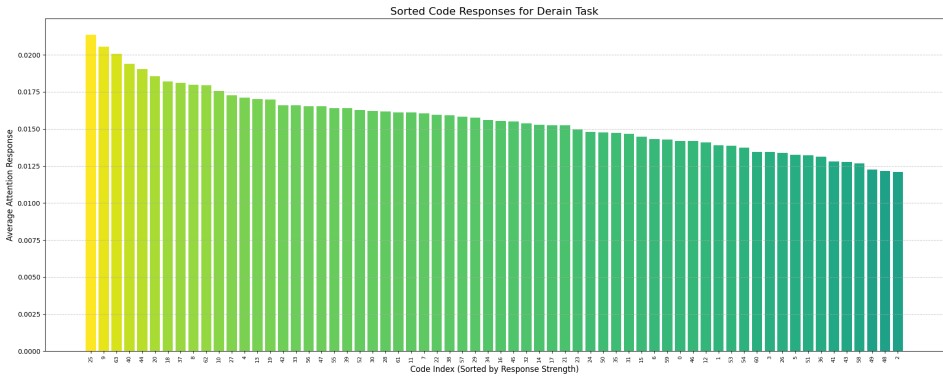

Figure 8: The complete distribution of the 64 code primitives' responses for the **Deraining** task on the Rain100L dataset.

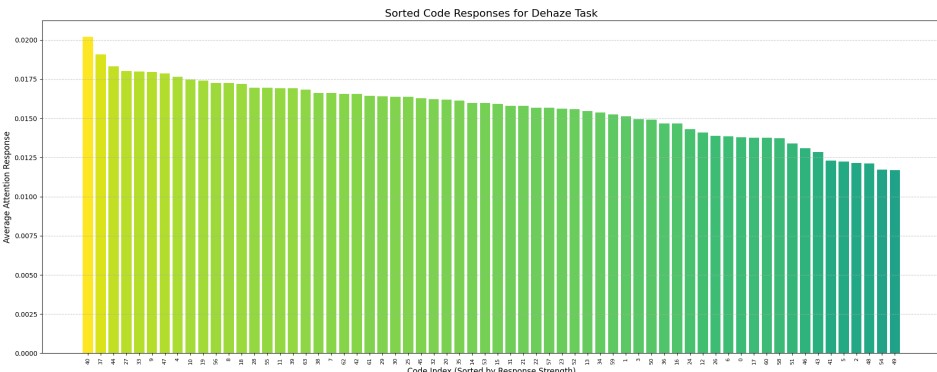

Figure 9: The complete distribution of the 64 code primitives' responses for the **Dehazing** task on the SOTS dataset.

to generate a channel-wise attention map of size $C \times C$. This map represents the inter-dependencies between different feature channels. The attention map is then applied to the value projection $V$ to produce the aggregated features. Finally, the output is passed through a $1 \times 1$ convolution and added to the input $X_{in}$ via a residual connection to form the output $X_{out}$.

**Gated-Dconv Feed-Forward Network (GDFN)** The GDFN, which we refer to as the FFN throughout our paper, serves as the primary feature transformation unit within each block. As illustrated in Figure 6(c), it replaces the standard FFN found in conventional Transformers with a more powerful gating mechanism.

Given a layer normalized input $X_{in}$, the GDFN first splits the processing into two parallel branches. The first branch applies a $1 \times 1$ convolution followed by a $3 \times 3$ depth-wise convolution and a GELU activation function to process the features. The second branch follows a similar convolutional path but without a non-linear activation, effectively acting as a learnable gate. The outputs of these two branches are then fused via element-wise multiplication. This gating mechanism allows the network to control which information is propagated forward, enhancing its representational power. Finally, the gated features are passed through a final $1 \times 1$ convolution, and the result is added back to the input $X_{in}$ via a residual connection.

## A.3 FULL CODEBOOK ACTIVATION VISUALIZATIONS

This section provides the complete, unabridged visualizations of the codebook activation responses for each of the three primary restoration tasks discussed in the main paper. While the main paper presents condensed visualizations (e.g., activation fingerprints and Top-K charts) for brevity and

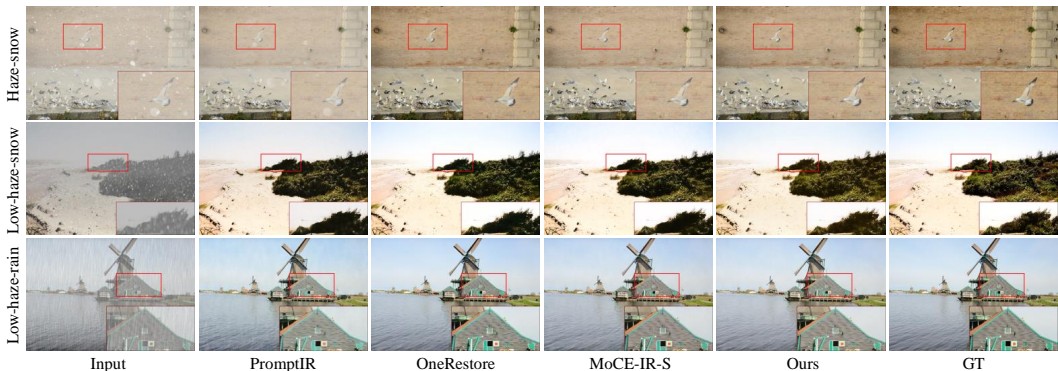

Figure 10: **Qualitative comparison on challenging composite degradations from the CDD11 dataset.** The examples shown are (*from top to bottom*): haze-snow, low-light-haze-snow, and low-light-haze-rain.

clarity, these full-distribution plots allow for a detailed inspection of the entire 64-primitive codebook's behavior. The results shown here correspond to the activations from the DACode module at Layer 1.

The activation patterns shown in Figure 7, 8, and 9 provide a comprehensive view of the functional specialization within our codebook. While each task has a unique activation signature, the full distributions allow for deeper analysis. For instance, one can observe the long tail of less-activated primitives and how their relative importance shifts between tasks. These plots serve as the foundational data from which the more condensed visualizations and analyses in the main paper are derived.

### A.4  ADDITIONAL QUALITATIVE RESULTS

To further demonstrate the superior performance and robustness of our proposed DACode framework, this section provides additional qualitative results. We present a wider variety of challenging visual examples for the Denoising, Deraining, and Dehazing tasks, comparing our method against several state-of-the-art approaches. As shown in Figure 11, the columns compare results from different methods against our proposed DACode and the Ground Truth. In the **Denoising** examples, our method excels at restoring intricate textures and natural colors (e.g., the tiger's fur and the pattern on the book cover), avoiding the color shifts or blurriness present in other methods. For **Deraining**, DACode effectively removes heavy rain streaks while preserving challenging background details (e.g., the jets on the tarmac and the soldiers' uniforms). In the **Dehazing** scenarios, our approach recovers significantly more vibrant colors and sharper text details (e.g., the "STATION" sign ), outperforming competitors that leave a residual hazy appearance.

To visually substantiate these strong quantitative results on composite degradations, we present challenging qualitative comparisons in Figure 10. A consistent pattern emerges across scenarios like haze-snow and low-light-haze-rain: while competing methods often struggle to handle the co-occurrence of multiple degradations—resulting in residual haze, color casts, or detail loss—our DACode-S demonstrates a superior ability to disentangle them. For instance, in the windmill scene (bottom row), DACode-S is the only method to effectively remove the thick atmospheric interference and restore the fine structural details, closely matching the ground truth. Similarly, in the beach scene (middle row), our approach excels at recovering the vibrant green hues of the foliage and correcting the severe color cast, whereas other methods yield washed-out or color-biased results. These visual results provide compelling, intuitive evidence for our compositional paradigm, highlighting its effectiveness in generating high-fidelity restorations for complex degradation mixtures.

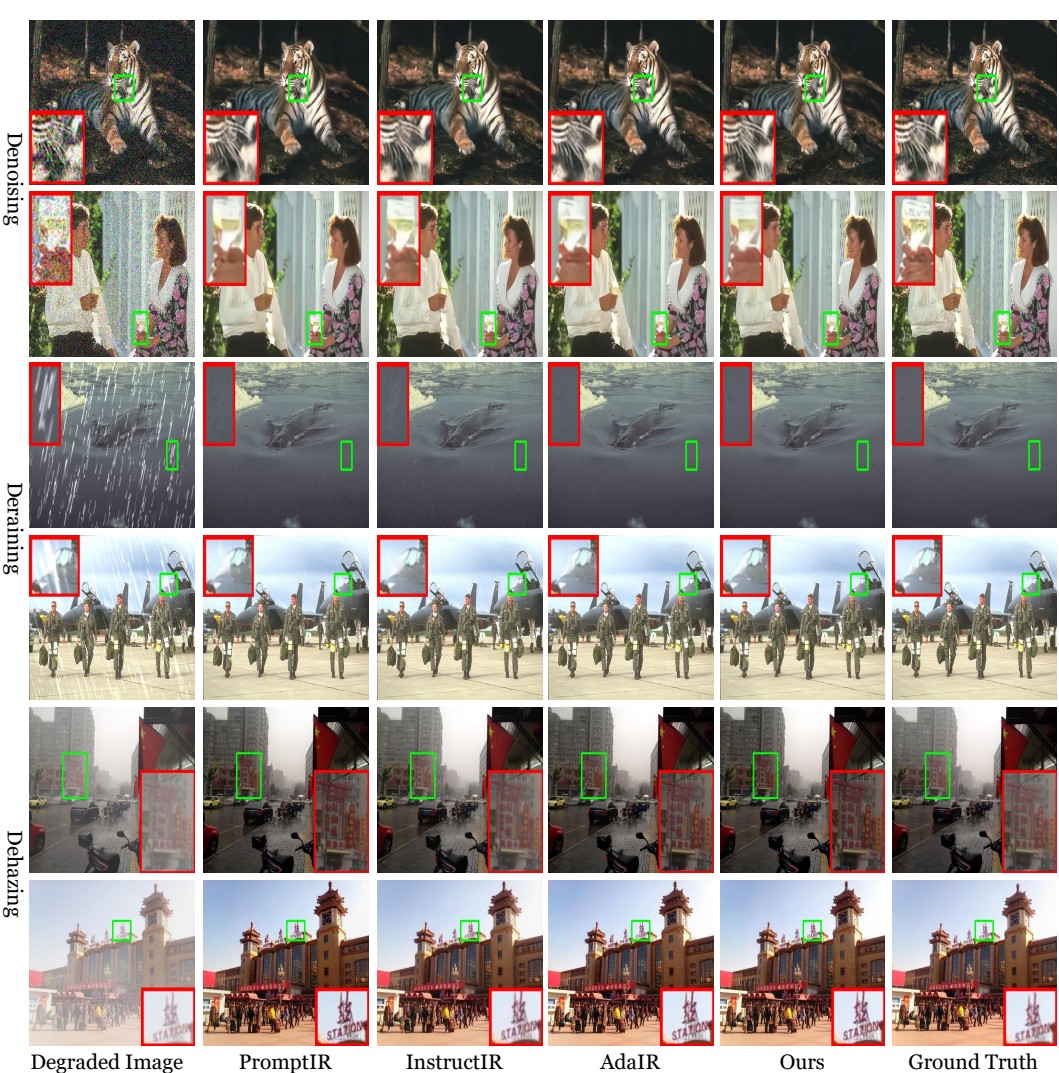

Figure 11: **Additional qualitative comparisons for all-in-one restoration on the 3-task benchmark.** The figure presents two additional challenging examples for each of the three tasks: **Denoising** (rows 1-2), **Deraining** (rows 3-4), and **Dehazing** (rows 5-6).

