# OpenReview forum: "Learning to Compose Degradations: A Codebook of Primitives for All-in-One Image Restoration"
_ICLR.cc/2026/Conference — ICLR 2026 Conference Withdrawn Submission_

### Official Review · Reviewer_khZo · 2025-10-23

**Soundness:** 3
**Presentation:** 3
**Contribution:** 2
**Rating:** 4
**Confidence:** 4

**Summary:**

The paper addresses the problem of all-in-one image restoration, arguing that existing categorical or prompt-based methods (e.g., treating degradations as discrete “haze” or “rain” types) fail to capture the continuous and compositional nature of real-world corruptions. It proposes DACode, a framework that represents degradations as mixtures of learnable primitives within a global codebook, with a dual cross-attention mechanism that adapts these primitives to input features and modulates restoration accordingly. The method reports state-of-the-art results on standard 3-/5-task and composite (CDD11) benchmarks.

**Strengths:**

1. The method replaces fixed task labels with a learnable codebook of primitives, allowing adaptive blending across degradation types and severities without separate prompts or heads.
2. A two-step interaction first adapts the codebook to the input and then uses it to guide restoration, promoting context-aware specialization.
3. The method shows solid results across multiple benchmarks.

**Weaknesses:**

1. The idea and motivation are not particularly novel. Using a codebook to represent and distinguish degradations has been explored in prior work, such as “Neural Degradation Representation Learning for All-In-One Image Restoration,” which is also an open-source method. As a result, the contribution of this paper appears to be incremental rather than fundamentally innovative.

2. The paper primarily presents average activation differences (e.g., Fig. 4), but the observed gaps are small and lack variance reporting, significance testing, or basic discriminative analyses. Without such probes or robustness checks, it remains unclear whether the learned primitives genuinely distinguish degradation types and severities or merely capture noisy correlations.

3. DACode is inserted only in the decoder, yet the encoder also shapes how degradations are captured and compressed. There’s no systematic study of alternatives (encoder-only, encoder+decoder, different layers/counts), so we don’t know if the current placement is optimal.

**Questions:**

1. In Fig. 1, it is difficult to visually distinguish the difference between the two images labeled [PSNR: 27.3 dB] and [PSNR: 35.6 dB]; they appear almost identical, as if one were a copy of the other.

---

### Official Review · Reviewer_A2a8 · 2025-11-01

**Soundness:** 2
**Presentation:** 2
**Contribution:** 2
**Rating:** 4
**Confidence:** 4

**Summary:**

This paper proposes DACode, a new framework for all-in-one image restoration. It challenges the conventional approach of representing degradations as discrete categories and instead posits that degradations are compositional, arising from a finite set of learnable "primitives" stored in a global codebook. The core mechanism is a two-stage, dual cross-attention process where the codebook primitives first query the input image features to become context-adapted , and subsequently, the image features query this adapted codebook to perform feature modulation and restoration. The method achieves strong state-of-the-art results on several benchmarks, including those for composite degradations

**Strengths:**

The conceptual shift towards a compositional, fine-grained degradation model is well-motivated and intuitive , and the model achieves impressive state-of-the-art empirical performance, particularly on complex composite degradation benchmarks.

**Weaknesses:**

My primary concern is the methodological novelty. The core technical contribution, the two-stage dual cross-attention module, appears nearly identical to the mechanism in "Prompt-In-Prompt Learning for Universal Image Restoration" (PIP) and shares significant overlap with "Boosting All-in-One Image Restoration via Self-Improved Privilege Learning." Specifically, the "Context-Aware Code Adaptation" stage  (where codebook queries image features) and the "Code-based Feature Modulation" stage  (where image features query the adapted codebook) directly mirror the prompt-in-prompt design. The paper frames this as a 'codebook' of 'primitives,' but the implementation is essentially a re-branding of an existing prompt-learning architecture. This significantly diminishes the paper's contribution, and the authors must explicitly discuss and differentiate their work from these prior arts.

Secondly, the paper does not adequately address the computational and training efficiency. Inserting this dual-attention module into each decoder stage adds non-trivial overhead, but the analysis is limited to parameter counts  without FLOPs or latency comparisons.

Finally, the paper's core claim of a fine-grained, compositional model that should generalize better is not fully substantiated. The evaluation is limited to synthetic benchmarks . There is no validation on real-world, in-the-wild degradation datasets. It is unclear if these 64 'primitives'  have learned fundamental degradation properties or have simply overfit to the specific synthetic degradation types in the training mix. A cross-domain or zero-shot experiment on a completely unseen, real-world dataset is necessary to validate the ambitious claims of learning a "rich, internal language of degradation".

**Questions:**

Can the authors precisely articulate the architectural differences between the proposed DACode module  and the core mechanism in PIP? If the architectures are indeed identical, the justification for this work's methodological contribution must be clarified.

Furthermore, given the visualizations in Figures 4 and 5 , what happens when the model is presented with a completely unseen degradation type (e.g., JPEG compression artifacts) or a novel combination of tasks not seen in the training set?

---

### Official Review · Reviewer_5a1R · 2025-11-01

**Soundness:** 2
**Presentation:** 2
**Contribution:** 2
**Rating:** 2
**Confidence:** 5

**Summary:**

This paper proposes a new CNN-based method to tackle all-in-one image restoration problem. In this paper, the all-in-one context focuses on bad weather conditions including dehazing, deraining, de-snowing, as well as denoising tasks. The proposes method introduces a code book vector which is a learnable vector. The code book vector is embedded into attention structure twice to categorically encode and split the degradation types. Extensive experiments show that the proposed method outperforms baseline methods on the suggested benchmarking datasets in dehazing, deraining, deblurring, low-light enhancement and denoising tasks.

**Strengths:**

1. The proposed method outperforms the baseline methods in each of the image enhancement task shown in the quantitative benchmarks.

**Weaknesses:**

1. This paper is not well written and revised. There is even a typo in the title. "Restoratio -> Restoration"
2.  It is not stated clearly that if the evaluated model is simultaneously trained on all the datasets of different task types in Sect 4.2 implementation details. The main all-in-one framework's goal is to conduct training process only once and apply the model for multiple tasks.
3. For deraining tasks, the main paper and supplementary materials are both showing synthetic rain images. The authors are suggested to demonstrate real-world deraining results to verify the effectiveness of the proposed method.
4. Figure 3, dehazing task does not show obvious superiority of the proposed method.
5. One of the most concerning points for the proposed method is about the theoretical relationship between the proposed new codebook and image restoration task types. It is not explicitly described why the codebook vector may modularize the input image features based on the degradation types automatically.
6. The novelty of the proposed method is quite limited as there are a number of codebook search algorithms for image restoration problems.

**Questions:**

The authors are suggested to re-polish the paper carefully and address the main issue from the weaknesses section during rebuttal period. The  main concern is regarding the theoretical reasoning of the codebook and the degradation types.

---

### Note · Authors · 2026-01-09

I have read and agree with the venue's withdrawal policy on behalf of myself and my co-authors.